# Generalized and Invariant Single-Neuron In-Vivo Activity Representation Learning

**Wei Wu[1], Yuxing Lu[1], Zhengrui Guo[2], Chi Zhang[1], Can Liao[3], Yifan Bu[1], Fangxu Zhou[1‡], Jinzhuo Wang[1‡]**

[1] Peking University, Beijing, China
[2] Hong Kong University of Science and Technology, Hong Kong, China
[3] University of Georgia, Athens, USA

‡ Corresponding author: `wangjinzhuo@pku.edu.cn`, `zhoufangxu@pku.edu.cn`

## Abstract

In neuroscience, models that learn representations of single-neuron in-vivo activity are essential for understanding the functional identities of individual neurons. The primary goal of these models—spanning Transformer-based, contrastive, and variational autoencoder frameworks, is not to predict neural activity, but to distill it into a stable, low-dimensional embedding that captures a neuron's intrinsic features. These learned identity embeddings should be invariant to changing experimental conditions while reflecting the neuron's molecular type and anatomical location, thus enabling downstream tasks like in-vivo cell type prediction. However, current models suffer from limited generalizability due to batch effects: non-biological variations arising from differences in experimental design, animal subjects, or recording platforms. These batch effects cause overfitting, reducing model robustness and utility. Crucially, previous work has not rigorously evaluated model performance on unseen, or "out-of-domain," animals and stimuli, creating a significant gap in the field. To solve this, we first introduce a comprehensive benchmark protocol that explicitly evaluates generalization to unseen batches. Second, we propose a model-agnostic adversarial training strategy where a discriminator network forces the primary model to learn embeddings that are invariant to batch information. Our approach is compatible with all major single-neuron representation models and significantly improves their robustness. This work highlights the critical need for generalization in such models and offers an effective solution, paving the way for the creation of unified neural atlases from in-vivo activity.

## 1 Introduction

In the field of neuroscience, the exploration of the intricate mechanisms underlying neural activity has been an enduring endeavor. A key challenge is to understand the stable, functional identities of individual neurons from their dynamic activity patterns. To this end, single-neuron representation models have emerged as a crucial tool [1–4]. The central goal of these models is not to predict moment-to-moment neural activity, but rather to distill the complex activity of a neuron into a stable, low-dimensional embedding that represents its intrinsic functional identity. This learned identity should be independent of transient factors like the specific stimuli an animal receives, and is instead related to fundamental properties such as the neuron's molecular type, anatomical position, and connectivity status.

There are three main data-driven paradigms for learning these identity embeddings. The first encompasses models featuring implicit representations, typically built on Transformer architectures, which elegantly define the representation of each neuron as a learnable identity vector, as elaborated in [5, 6].

39th Conference on Neural Information Processing Systems (NeurIPS 2025).

The second paradigm centers on contrastive learning, which learns a robust embedding by maximizing the similarity of representations derived from different modalities of a single neuron's activity (e.g., its waveform and firing patterns) [7]. The third paradigm relies on variational autoencoders (VAEs), which learn a compressed latent embedding by training on an activity reconstruction task [8–10].

The primary output of these models is a learned feature vector—the identity embedding—that encapsulates the neuron's time-invariant and intrinsic properties. The utility of this embedding is then validated by its performance on downstream tasks, such as facilitating the prediction of molecular cell types and anatomical locations for a single neuron. These predictive capabilities are critical for enabling advanced, long-term closed-loop experiments, for instance, by allowing for targeted perturbations of specific, functionally-identified neurons [11].

However, existing single-neuron representation models face significant challenges in terms of generalizability. Batch effects (non-biological variations arising from different experimental designs or animal subject) can contaminate the learned embeddings (Figure 1 a). These batch effects cause models to overfit to specific experimental conditions, severely compromising their ability to generalize. For instance, an embedding model trained on data from one animal may fail on another, even for the same cell type in the same brain region. Previous studies have often overlooked this issue, failing to rigorously evaluate model generalization on new animals and stimuli (out-of-domain). This overlooks the critical aspect of model robustness, which is essential for the practical application of these models in broader neuroscience research.

To address these issues, this paper introduces a comprehensive benchmark and a robust training strategy. In the following sections, we will present a series of experiments, such as cell-type prediction across different stimuli or across different animals. For each experiment, we will first detail the specific setup and protocol, and then immediately present the results, providing a clear and self-contained evaluation of model generalization. This structure is designed to rigorously assess model performance on previously unseen data, filling a key gap in prior research.

Moreover, we propose a model-agnostic strategy (Figure 1 b) to mitigate the influence of batch effects. Adversarial training has proven effective in various machine learning domains for removing unwanted variations [12]. In our context, an adversarial discriminator attempts to predict batch information from the learned neuron embeddings. In response, the primary model is trained to produce embeddings that are invariant to the batch identity—meaning the representations are statistically indistinguishable regardless of which animal or stimulus condition they came from. This forces the model to focus on intrinsic biological features rather than experimental artifacts. Our framework is designed to be compatible with all three major representation learning paradigms, enhancing their generalizability.

In summary, this study underscores the significance of generalization in single-neuron representation models. By leveraging this training, we aim to enhance the robustness of the learned identity embeddings, mitigating batch effects. This approach not only advances the development of single-neuron representation models but also paves the way for their broader practical applications in neuroscience research.

## 2 Related Work

**NeuPRINT-Implicit Single Neuron Representations [5]:** The pursuit of implicit single neuron representations has been driven by the recognition that the in vivo physiology of a neuron comprises two distinct elements: the neuron's inherent properties and the synaptic and modulatory inputs it receives. To disentangle these components, recent studies have proposed self-supervised approaches for inferring identity vectors for neurons. These methods rely on models of neuronal dynamics with exogenous inputs, leveraging the temporal structure of neural activity to extract neuron-specific features. For instance, by formulating a model of activity dynamics that depends solely on a neuron's past activity and population-level statistics invariant to individual ordering, researchers have been able to infer neuronal identities from population recordings. This approach capitalizes on the assumption that the underlying dynamics of each neuron, despite being influenced by external inputs, possess unique signatures that can be captured through self-supervision. These implicit representations offer a powerful way to summarize a neuron's functional characteristics without explicitly defining its features, but they may still be affected by factors such as batch-specific variations in experimental conditions.

**NEMO-Multimodal Contrastive Learning [7]:** Multimodal contrastive learning has emerged as a promising strategy for single neuron representation learning. By jointly embedding different

modalities of single-neuron data, such as activity autocorrelations and waveforms, these methods aim to capture the shared information across modalities while discarding modality-specific noise. The underlying assumption is that neurons with similar functional roles will exhibit similar patterns across multiple modalities, and by maximizing the similarity of their representations in a shared latent space, a more robust and discriminative representation can be obtained. For example, recent studies have utilized advanced contrastive learning frameworks to map these diverse modalities into a common embedding space, effectively enhancing the representational power of single-neuron models. This approach not only leverages the complementary information from multiple sources, but also helps in reducing the impact of noise within individual modalities. However, like other methods, it remains vulnerable to batch effects, which can introduce confounding variations that distort the learned representations and limit their generalizability.

**VAE-Based Method [10]:** VAEs have also been widely employed for unsupervised pre-training in single-neuron representation learning. VAEs compress high-dimensional input data, such as neuronal waveforms and activity autocorrelations, into lower-dimensional latent spaces. Through the optimization of reconstruction loss and KL-divergence, VAEs learn to represent the essential features of single neuron data while discarding redundant information. These learned latent representations provide a compact summary of single neuron activity patterns, which can be further utilized for tasks such as cell type classification and brain region prediction.

**End2End Method:** End-to-end supervised training methods are the common approaches for predicting cell types/brain regions directly based on individual neuronal activities. We have adopted the well-established LOLCAT as the end-to-end basemodel [13]. The end-to-end method attempts to directly extract and map molecular type labels or anatomical location information from neuronal activities. However, it cannot comprehensively represent the functional identity information of neurons in advance like the self-supervised method. The prediction of labels is merely a verification and partial application of the learned functional identity representation.

Adversarial strategies for eliminating batch effects have been used in the field of single-cell omics before [14]. To our knowledge, we are the first to treat in vivo activity as an omics and apply adversarial strategies, but the goal of this paper is to demonstrate the effectiveness of adversarial strategies. In-depth comparison and selection of adversarial methods is a promising future work.

## 3 Method

We introduce a model-agnostic adversarial training framework to mitigate batch effects in single-neuron representations, thereby improving generalization across experimental conditions and animal subjects. Our approach is compatible with mainstream single-neuron representation paradigms, including implicit Transformer-based models, contrastive learning, and VAE).

### 3.1 Problem Formulation

Let $\mathcal{X} = \{\mathbf{x}_i\}_{i=1}^N$ be raw neuronal observations, each associated with an intrinsic label $y_i \in \mathcal{Y}$ (e.g., cell type, region) and a batch label $\mathbf{b}_i \in \mathcal{B}$ (e.g., animal ID, stimulus, session). The objective is to learn an encoder $f_\theta : \mathcal{X} \to \mathbb{R}^d$ that maps $\mathbf{x}_i$ to an embedding $\mathbf{z}_i = f_\theta(\mathbf{x}_i)$, such that $\mathbf{z}_i$ preserves $y_i$-relevant features while being invariant to batch effects. Standard models often overfit to batch-specific information, resulting in poor generalization to unseen batches.

### 3.2 Adversarial Training Framework

Our framework consists of two modules: (1) a representation encoder $f_\theta$ that extracts embeddings $\mathbf{z}_i$, and (2) a batch discriminator $D_\phi$ that predicts batch labels from $\mathbf{z}_i$. Training is adversarial: $D_\phi$ is optimized to identify batch information, while $f_\theta$ is trained to remove it, encouraging batch-invariant representations.

### 3.3 Objective Functions

The total loss combines a task-specific base loss and an adversarial loss:

- **Base Loss $\mathcal{L}_{\text{base}}$:** Defined by the specific single-neuron representation paradigms, e.g., cross-entropy for classification (LOLCAT), reconstruction loss for VAE, or contrastive loss for self-supervised learning (NEMO).
- **Adversarial Loss $\mathcal{L}_{\text{batch}}$:** Cross-entropy loss for batch prediction. $D_\phi$ minimizes this loss, while $f_\theta$ maximizes it to confuse the discriminator.

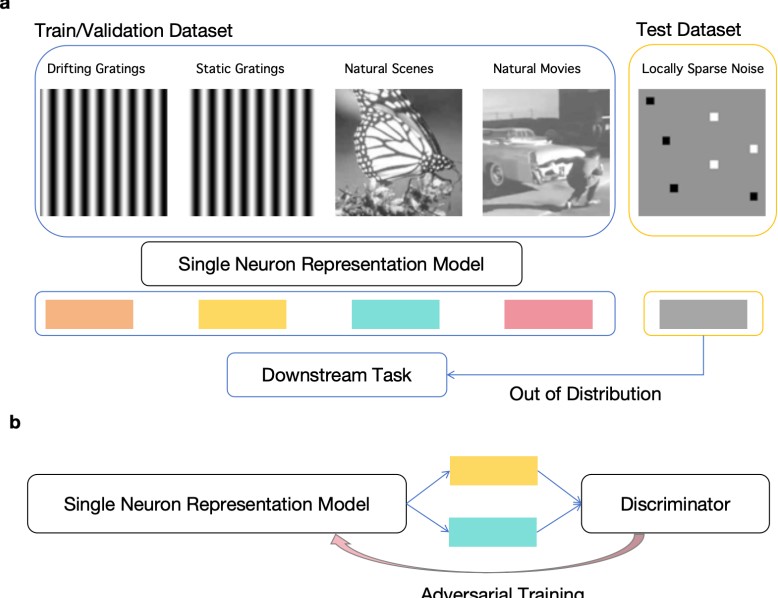

Figure 1: Schematic overview of the experimental protocol and adversarial training framework for evaluating and improving single-neuron representation models. (a) Illustration of the experimental protocol designed to rigorously assess the generalization ability of single-neuron representation models. The training and validation dataset comprises diverse visual stimuli, including drifting gratings, static gratings, natural scenes, and natural movies, which are used to train the single-neuron representation model. The learned representations are subsequently evaluated on a downstream task. To assess out-of-distribution generalization, a distinct test dataset consisting of locally sparse noise stimuli is employed, which is not present in the training/validation set. This protocol enables explicit evaluation of model performance on novel stimulus conditions, providing a stringent test of generalizability. (b) Schematic of the model-agnostic adversarial training framework. Detail in Method Section and Figure 2.

The joint objective is:

$$\min_{\theta} \max_{\phi} \quad \mathcal{L}_{\text{base}}(f_\theta) - \lambda \mathcal{L}_{\text{batch}}(D_\phi(f_\theta)),$$

where $\lambda$ controls the adversarial regularization strength. We employ a Gradient Reversal Layer (GRL) [12] to enable efficient end-to-end optimization.

### 3.4   Model-Agnostic Integration

Our framework is architecture-agnostic: $f_\theta$ can be any single-neuron activity encoder (e.g., implicit transformer, contrastive based MLP, autoencoder), and $D_\phi$ is a two hidden layer MLP operating on $\mathbf{z}_i$. Hyperparameters are selected via cross-validation.

### 3.5   Intuitive Explanation

Our adversarial approach is intuitively illustrated by figure 2. Without adversarial training, a model can be confounded by batch effects (colors), learning a decision boundary that is not based on true biological features (shapes) and thus fails to generalize to unseen test data (left panel). Adversarial training corrects this by forcing the model to create batch-invariant embeddings that are indistinguishable to a discriminator network. As a result, the model is compelled to learn a more robust and generalizable decision boundary based only on the intrinsic features of the cell types, leading to correct classification on novel data (right panel). This process aligns with the principle of invariant risk minimization, ensuring the model relies on biologically meaningful signals rather than experimental artifacts.

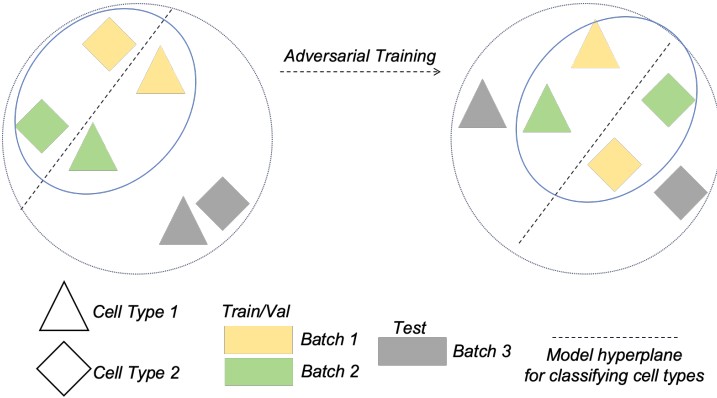

Figure 2: Illustration of adversarial training for improving the generalizability of single-neuron representation models. Each shape represents a neuron from a specific cell type, and colors indicate different experimental batches. Left: Without adversarial training, the model hyperplane (dashed line) for cell type classification is confounded by batch effects, leading to poor generalization on unseen batches. Right: Adversarial training encourages the model to learn batch-invariant representations, resulting in a more robust and generalizable decision boundary across batches. This approach enables reliable cell type classification even in novel experimental conditions.

## 4    Benchmark Experiment Setup

We focus on evaluating the impact of adversarial training on the performance of the original model. Detail implementation can be found in appendix  B LOLCAT, C NeuPRINT, D VAE. Key implementation details (optimizer, schedule, discriminator capacity) sit in Appendix A.5.

### 4.1    Cell Type Prediction Across Visual Stimulus

For this task, we use the V1-CellType dataset.

**V1-CellType [15]:** This dataset is from multimodal recordings of the mouse primary visual cortex (V1). 2-photon calcium imaging with a 4.3Hz temporal sampling frequency was used to obtain population activity recordings, and the spatial coordinates of recorded neurons were provided. Functional recordings were obtained from four mice, namely SB025, SB026, SB028, and SB030. Recordings were perfomed under 4 sets of visual stimuli. In total, these recordings encompass 9728 neurons.  Each session endured approximately 20 minutes and documented around 500 neurons. Subsequently, single cell spatial transcriptomics was conducted on the tissue. The mRNA expression of 72 selected genes in ex-vivo tissue was profiled to classify neurons as excitatory or inhibitory. Neurons were further subclassified into Lamp5, Pvalb, Vip, and Sst subtypes.

We conduct experiments on this dataset to evaluate cell type prediction across the four distinct visual stimulus conditions: spontaneous activity, drifting gratings, and two different natural scene image sets. Previous studies have typically either pooled data from all stimulus conditions and performed random splits for training and validation, or focused on a single stimulus type. In contrast, our experimental protocol is designed to rigorously assess the generalization ability of models to novel stimulus conditions. Specifically, we train and validate both the representation model and the cell type classifier using neurons recorded under three of the four stimulus conditions, and then evaluate the model on neurons recorded under the remaining, held-out stimulus condition. This out-of-domain evaluation protocol ensures that the model is tested on data distributions not encountered during training, providing a stringent assessment of its robustness and generalization to unseen stimuli.

We systematically compare model performance under three experimental settings: (i) without adversarial training and without splitting by stimulus condition (i.e., neurons from all stimulus conditions are mixed and randomly split), (ii) without adversarial training but with splitting by stimulus condition (cross-stimulus evaluation), and (iii) with adversarial training under the cross-stimulus evaluation protocol. This comprehensive comparison allows us to quantify the impact of adversarial training on model robustness and generalization across different stimulus conditions. As with the cross-animal experiments, we do not evaluate the effect of adversarial training on the NEMO approach for this

dataset due to the complexity of preprocessing the V1-CellType dataset into the multimodal input format required by the NEMO model.

## 4.2 Cell Type Prediction Across Animals

We continue to conduct experiments on the V1-CellType dataset to assess cell type prediction across animals. Previous studies have either performed experiments on individual animals or mixed neurons from all animals together before randomly splitting them into training and validation sets. As described above, the V1-CellType dataset contains recordings from four mice. We train and validate both the representation model and the cell type classifier using neurons from three mice, and then evaluate the model on neurons from the remaining mouse, which is held out entirely from model optimization. This out-of-domain testing protocol allows us to rigorously assess the generalization ability of the learned representations and classifiers to unseen animals.

We systematically compare the model performance under three conditions: (i) without adversarial training and without splitting by animal ID (i.e., neurons from all animals are mixed and randomly split), (ii) without adversarial training but with splitting by animal ID (cross-animal evaluation), and (iii) with adversarial training under the cross-animal evaluation protocol. This enables us to quantify the effect of adversarial training on model robustness and generalization across animals. Due to the challenges in preprocessing the V1-CellType dataset into the multimodal input format required by the NEMO model, we do not evaluate the effect of adversarial training on the NEMO approach for this dataset.

## 4.3 Anatomical Brain Region Prediction Across Animals

To further evaluate the adversarial training, we conduct experiments on the IBL Brain-wide Map dataset, focusing on anatomical brain region prediction across animals.

**IBL Brain-wide Map [16]:** This dataset is a comprehensive set of recordings from 115 mice performing a decision-making task with sensory, motor, and cognitive components, obtained with 547 Neuropixels probe insertions covering 267 brain areas in the left forebrain and midbrain and the right hindbrain and cerebellum. Following recordings, probe tracks were reconstructed using serial-section 2-photon microscopy, and each recording site and neuron was assigned a region in the Allen Common Coordinate Framework. Annotations divided into 10 broad areas by Cosmos hierarchical grouping: isocortex, olfactory areas (OLF), cortical subplate(CTXsp), cerebral nuclei (CNU), thalamus (TH), hypothalamus (HY), midbrain (MB), hindbrain (HB), cerebellum (CB) and hippocampal formation (HPF).

In previous studies using this data, models have often been trained and validated on data pooled from multiple animals, with random splits that do not account for inter-animal variability. In contrast, our experimental protocol is designed to rigorously assess the ability of models to generalize anatomical region predictions to entirely unseen animals. Specifically, we train and validate both the representation model and the anatomical region classifier using neurons recorded from a subset (0.8) of animals, and then evaluate the model on neurons from a held-out animal that is excluded from all stages of model optimization. This out-of-domain evaluation protocol ensures that the model is tested on data distributions that reflect biological variability across individual animals, providing a stringent assessment of its robustness and generalization.

We systematically compare model performance under three experimental settings: (i) without adversarial training and without splitting by animal subjects (i.e., neurons from all animals are mixed and randomly split), (ii) without adversarial training but with splitting by animal subjects (cross-animal evaluation), and (iii) with adversarial training under the cross-animal evaluation protocol. This comprehensive comparison enables us to quantify the impact of adversarial training on model robustness and generalization across animals in the context of anatomical brain region prediction. We do not evaluate the effect of adversarial training on the NeuPRINT approach for the IBL Brain-wide Map dataset due to the complexity of preprocessing required for NeuPRINT input formats.

## 5   Result

**Cell Type Prediction Across Visual Stimulus (V1-CellType)**

**Cell Type Prediction Across Visual Stimulus (V1-CellType)** In Table 1, we evaluate the performance of different models on cell type prediction under various visual stimulus conditions. The results reveal distinct patterns of model behavior across the three experimental settings. The NeuPRINT

Table 1: Cell Type Prediction Across Visual Stimulus (V1-CellType). Each cell shows the top-1 accuracy. The "Out-of-domain" columns show the performance on the generalization task and, in parentheses, the percentage decrease relative to the "In-domain" result.

| Model | In-domain (Random Split) | Out-of-domain (Cross-Stimulus) | Out-of-domain (+Adversarial) |
|---|---|---|---|
| LOLCAT (End2End) | 0.746 | 0.358 (-52.0%) | 0.566 (-24.1%) |
| NeuPRINT (Implicit) | 0.787 | 0.644 (-18.2%) | 0.743 (-5.6%) |
| VAE (Explicit) | 0.732 | 0.569 (-22.3%) | 0.669 (-8.6%) |

(Implicit) model demonstrates the strongest baseline performance with an in-domain accuracy of 0.787, followed by LOLCAT (End2End) at 0.746 and VAE at 0.732. However, when tested under cross-stimulus conditions, all models experience significant performance degradation, with LOLCAT showing the most dramatic drop (-52.0%) to 0.358, while NeuPRINT maintains relatively better performance with a more modest decrease (-18.2%) to 0.644. The introduction of adversarial training leads to substantial improvements across all models, with LOLCAT showing the most remarkable recovery (+58.1%) to 0.566. This suggests that while End2End is more sensitive to stimulus variations, it benefits the most from adversarial training in terms of relative improvement. NeuPRINT, despite its initial robustness, still shows meaningful improvement (+15.4%) with adversarial training, achieving the highest final accuracy of 0.743.

Table 4, Table 5, and Table 6 provide a detailed breakdown of model performance when each visual stimulus condition is held out as the test set. Across all models, we observe a consistent trend: prediction accuracy is lowest when spontaneous activity is used as the test condition, moderately higher for drifting gratings, and highest for both natural scene image sets. This pattern highlights the particular challenge posed by spontaneous activity, which appears to be the most distinct from the other stimulus types in terms of neural response patterns.

Interestingly, we believe that the relatively poor performance on spontaneous and drifting grating conditions is not only due to their distributional differences, but also because these stimulus types are relatively simple and less diverse. As a result, neuronal activity under these conditions may not fully show the their computational roles.

Table 2: Cell Type Prediction Across Animals (V1-CellType). Each cell shows the top-1 accuracy. The "Out-of-domain" columns show the performance on the generalization task and, in parentheses, the percentage decrease relative to the "In-domain" result.

| Model | In-domain (Random Split) | Out-of-domain (Cross-Animals) | Out-of-domain (+Adversarial) |
|---|---|---|---|
| LOLCAT (End2End) | 0.746 | 0.452 (−39.4%) | 0.694 (−7.0%) |
| NeuPRINT (Implicit) | 0.787 | 0.582 (−26.0%) | 0.739 (−6.1%) |
| VAE (Explicit) | 0.732 | 0.557 (−23.9%) | 0.668 (−8.7%) |

**Cell Type Prediction Across Animals (V1-CellType)** Table 2 presents the performance of models on cell type prediction across different animals using the V1-CellType dataset. The results reveal interesting patterns in model generalization capabilities. Similar to the cross-stimulus scenario, all models show significant degradation under cross-animal conditions. LOLCAT experiences the most severe performance drop (-39.4%) to 0.452, while VAE shows relatively better preservation of performance with a -23.9% decrease to 0.557. The application of adversarial training leads to substantial improvements across all models, with LOLCAT again showing the most dramatic recovery (+53.5%) to 0.694. Notably, while NeuPRINT maintains the highest absolute accuracy throughout, LOLCAT demonstrates the most significant relative improvement with adversarial training, suggesting that adversarial training is particularly effective for models that are initially more sensitive to animal-specific variations.

**Anatomical Brain Region Prediction Across Animals (IBL Brain-wide Map)** In Table **??**, we assess the performance of different models on anatomical brain region prediction across animals using the IBL Brain-wide Map dataset. The results reveal a more balanced performance landscape compared to the V1-CellType experiments. All models show similar baseline performance, with

Table 3: Anatomical Brain Region Prediction Across Animals (IBL Brain-wide Map). Each cell shows the top-1 accuracy. The "Out-of-domain" columns show the performance on the generalization task and, in parentheses, the percentage decrease relative to the "In-domain" result.

| Model | In-domain (Random Split) | Out-of-domain (Cross-Animals) | Out-of-domain (+Adversarial) |
|---|---|---|---|
| LOLCAT (End2End) | 0.507 | 0.389 ($-23.1\%$) | 0.482 ($-4.9\%$) |
| NEMO (Contrastive) | 0.511 | 0.400 ($-21.7\%$) | 0.496 ($-2.9\%$) |
| VAE | 0.488 | 0.396 ($-18.9\%$) | 0.457 ($-6.4\%$) |

NEMO (Contrastive) slightly leading at 0.511, followed by LOLCAT at 0.507 and VAE at 0.488. Under cross-animal conditions, the performance degradation is more moderate compared to the V1-CellType experiments, with decreases ranging from -18.9% to -23.1%. The application of adversarial training leads to consistent improvements across all models, with NEMO showing the most significant relative improvement (+24.0%) to 0.496. Interestingly, while the absolute improvements are smaller than in the V1-CellType experiments, the relative improvements are more consistent across models, suggesting that adversarial training may be particularly effective for anatomical region prediction tasks. The final performance levels are also more closely clustered, with all models achieving accuracies between 0.457 and 0.496, indicating that this task may be inherently more challenging or that the models have reached a common performance ceiling.

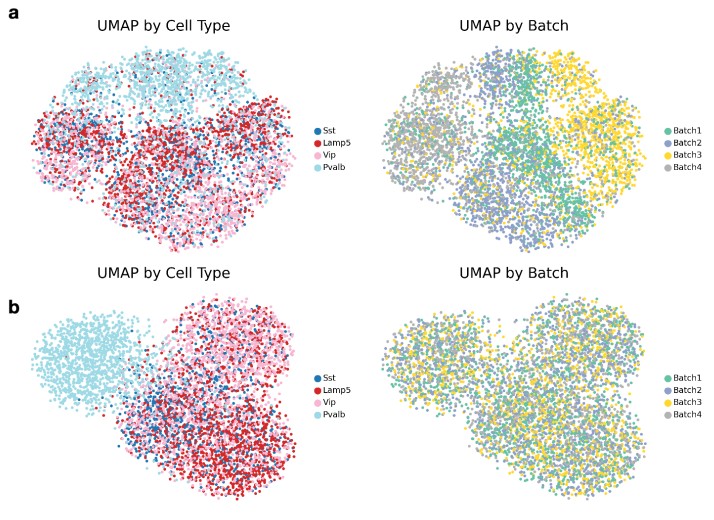

Figure 3: UMAP visualizations of single-neuron embeddings learned by the NeuPRINT model, colored by cell type (left) and batch/stimulus condition (right). Batch1 and Batch2 correspond to the two natural scene stimulus conditions (natural 01 and natural 02), while Batch3 and Batch4 correspond to spontaneous activity and drifting gratings, respectively. (**a**) Without adversarial training. (**b**) With adversarial training.

**Visualization** To qualitatively assess the impact of adversarial training on the batch invariance of single-neuron representations, we visualized the learned embeddings of the NeuPRINT model using UMAP, colored by both cell type and batch (stimulus condition), as shown in Figure 3. In the baseline setting without adversarial training (Figure 3a), the UMAP projection colored by cell type reveals partial separation between neuronal subtypes, indicating that the model captures some intrinsic cell type information. However, when colored by batch, the embeddings exhibit clear clustering according to the four stimulus conditions, demonstrating a pronounced batch effect. This suggests that the learned representations are confounded by stimulus-specific information, which may hinder generalization to novel conditions. In contrast, after applying our adversarial training framework (Figure 3b), the UMAP visualization shows that the separation between cell types is preserved or even enhanced, while the clustering by batch is substantially diminished. The embeddings corresponding to

different stimulus conditions are now well mixed, indicating that batch-related information has been effectively removed from the representations. This qualitative result demonstrates that adversarial training successfully disentangles intrinsic neuronal properties from batch effects, enabling the model to learn more robust and generalizable single-neuron representations. These findings are consistent with our quantitative results, highlighting the effectiveness of adversarial training in improving cross-condition generalization.

## 6 Disscussion

**Conclusion:** This work emphasizes the importance of generalization in single-neuron representation models for reliable real-world neuroscience applications. We established a rigorous evaluation protocol assessing performance on unseen animals and stimuli, offering a stringent robustness check. Our proposed model-agnostic adversarial training framework, which eliminates batch-related information from representations, consistently enhances cross-condition and cross-animal generalization across state-of-the-art models. This advancement promotes more robust practical use of such models in neuroscience research.

**Limitation:** Our adversarial training framework has notable limitations. First, experiments are restricted to mouse data, leaving cross-species generalization (e.g., rodents to primates/humans) untested—critical for translational research. Second, evaluations are limited to specific recording platforms and labs; batch effects from varying acquisition devices or protocols across labs pose unaddressed challenges for universal models. Third, some neuronal subtypes show poor cross-animal generalization post-training, potentially due to genuine biological variability or low evolutionary conservation. Additionally, comparisons with other batch-invariance methods (including non-adversarial and single-cell omics approaches) remain underexplored. Fourth, in this work, we did not treat different trials as distinct batches, which would require more complex experimental design. This is indeed a highly promising area for exploration, as some neurons exhibit reduced response strength upon repeated exposure to the same stimulus [17].

**Broad Impact:** At present, our study is limited to mouse datasets, and the broader societal impact is correspondingly constrained. However, the proposed adversarial training framework represents a significant step toward the development of more robust and generalizable single-neuron representation models. By mitigating batch effects and improving model generalization, our approach has the potential to facilitate the deployment of computational models in in vivo experiments, enabling more precise and reliable identification of neuronal cell types and anatomical locations. In the long term, this could accelerate the translation of computational neuroscience advances into experimental and clinical applications, ultimately contributing to a deeper understanding of brain function and disease.

## 7 Reproducibility

The process of reproducing these models was very difficult for us, and we made certain changes to the original code to ensure that it runs and converges. The original code can be accessed via the following links: NeuPRINT at `https://github.com/lumimim/NeuPRINT/`, LOLCAT at `https://github.com/nerdslab/lolcat`, and NEMO at `https://github.com/Haansololfp/NEMO_ICLR`. The data can be found at the links: IBL at `https://www.internationalbrainlab.com/brainwide-map` and V1-CellType at `https://figshare.com/articles/dataset/A_transcriptomic_axis_predicts_state_modulation_of_cortical_interneurons/19448531`. All work can be completed on a single A100.

## Acknowledge

This research was supported by National Key Research and Development Program of China (2024YFF0507400) and National Natural Science Foundation of China (6220071694).

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

# A    Detail Methods

In this section, we present our model-agnostic adversarial training framework designed to mitigate batch effects in single-neuron representations, thereby enhancing generalization across experimental conditions and animal subjects. Our approach integrates seamlessly with existing single-neuron representation paradigms, including Transformer-based models, contrastive learning frameworks, and variational autoencoders (VAEs).

## A.1    Problem Formulation

Let $\mathcal{X} = \{\mathbf{x}_i\}_{i=1}^N$ denote a collection of raw neuronal observations, where each $\mathbf{x}_i$ corresponds to a single neuron. Each neuron is associated with: - An intrinsic property label $y_i \in \mathcal{Y}$ (e.g., cell type, anatomical region), - A batch label $\mathbf{b}_i \in \mathcal{B}$ encoding experimental metadata (e.g., animal ID, stimulus condition, recording session).

The goal of single-neuron representation learning is to learn an encoder $f_\theta : \mathcal{X} \to \mathbb{R}^d$ that maps $\mathbf{x}_i$ to a low-dimensional embedding $\mathbf{z}_i = f_\theta(\mathbf{x}_i)$, such that $\mathbf{z}_i$ captures $y_i$-relevant intrinsic features while being invariant to $\mathbf{b}_i$-related batch effects. Downstream tasks (e.g., classification of $y_i$) are then performed using $\mathbf{z}_i$.

Key challenge: Standard models overfit to $\mathbf{b}_i$, leading to poor generalization when tested on neurons from unseen $\mathbf{b} \notin \text{support(training batches)}$.

## A.2    Adversarial Training Framework

Our framework consists of two components: 1.Representation Encoder $f_\theta$: Maps raw data $\mathbf{x}_i$ to embeddings $\mathbf{z}_i$, optimized to preserve intrinsic features relevant to $y_i$. 2.Batch Discriminator $D_\phi$: A classifier that takes $\mathbf{z}_i$ as input and predicts the batch label $\mathbf{b}_i$, optimized to detect batch-specific patterns in $\mathbf{z}_i$.

Training proceeds adversarially: $f_\theta$ aims to produce embeddings that "fool" $D_\phi$ (i.e., $\mathbf{z}_i$ contains no information about $\mathbf{b}_i$), while $D_\phi$ aims to accurately predict $\mathbf{b}_i$ from $\mathbf{z}_i$. This min-max game enforces $\mathbf{z}_i$ to be invariant to batch effects.

## A.3    Objective Functions

The total training objective combines a downstream task-specific base loss and an adversarial loss:

Base Task Loss $\mathcal{L}_{\text{base}}$ This loss depends on the specific representation learning paradigm and downstream task:

- Supervised/self-supervised learning: For classification tasks (e.g., cell type prediction), $\mathcal{L}_{\text{base}}$ is cross-entropy between predicted $y_i$ and ground truth.

- Reconstruction-based models (e.g., VAEs): $\mathcal{L}_{\text{base}}$ includes reconstruction loss (e.g., mean squared error) and latent distribution regularization.

- Contrastive learning: $\mathcal{L}_{\text{base}}$ is a contrastive loss maximizing similarity of augmentations from the same neuron and minimizing similarity across different neurons.

Adversarial Loss $\mathcal{L}_{\text{adv}}$ The discriminator $D_\phi$ is trained to minimize the batch classification loss $\mathcal{L}_{\text{batch}}(\mathbf{z}_i, \mathbf{b}_i)$, typically cross-entropy:

$$\mathcal{L}_{\text{batch}} = -\mathbb{E}_{(\mathbf{x}_i, \mathbf{b}_i) \sim \mathcal{D}} \left[ \log D_\phi(\mathbf{z}_i, \mathbf{b}_i) \right],$$

where $\mathcal{D}$ is the data distribution. The encoder $f_\theta$ is trained to maximize $\mathcal{L}_{\text{batch}}$ (i.e., confuse $D_\phi$), leading to the adversarial objective:

$$\mathcal{L}_{\text{adv}} = \mathbb{E}_{(\mathbf{x}_i, \mathbf{b}_i) \sim \mathcal{D}} \left[ \log(1 - D_\phi(\mathbf{z}_i, \mathbf{b}_i)) \right].$$

3.3.3 Joint Optimization The combined objective balances preserving task-relevant features and removing batch information:

$$\min_\theta \max_\phi \quad \mathcal{L}_{\text{base}}(f_\theta) - \lambda \cdot \mathcal{L}_{\text{batch}}(D_\phi \circ f_\theta),$$

where $\lambda > 0$ controls the strength of adversarial regularization. To simplify training, we use a Gradient Reversal Layer (GRL) [12], which applies a gradient sign reversal during backpropagation from $D_\phi$ to $f_\theta$, enabling end-to-end optimization without explicit min-max alternation.

## A.4 Model-Agnostic Architecture Design

Our framework is compatible with diverse single-neuron representation models by treating $f_\theta$ as a pluggable component:

- Transformer-based models (e.g., [5]): $f_\theta$ is a Transformer encoder that processes time-series spike data, with $\mathcal{L}_{\text{base}}$ defined by downstream classification or self-supervised objectives.

- Contrastive learning frameworks (e.g., [7]): $f_\theta$ encodes multimodal inputs (waveform, 3D ACG), and $\mathcal{L}_{\text{base}}$ includes contrastive losses across modalities, with $D_\phi$ operating on the unified embedding space.

- Variational autoencoders (e.g., [8]): $f_\theta$ is the VAE encoder, $\mathcal{L}_{\text{base}}$ combines reconstruction loss and KL divergence, and $D_\phi$ regularizes the latent space to be batch-invariant.

The discriminator $D_\phi$ is a multi-layer perceptron (MLP) with architecture [embedding dimension $d \to h_1 \to h_2 \to |\mathcal{B}|$], where $h_1, h_2$ are hidden layers.

## A.5 Implementation Details

For all experiments, models were trained using the Adam optimizer with an initial learning rate of 1e-4. The learning rate was reduced by a factor of 0.1 if the validation loss plateaued for 5 consecutive epochs. We used a batch size of 512 and trained with an early stopping criterion based on validation accuracy to prevent overfitting. The discriminator was implemented as a two-hidden-layer MLP with 256 units per layer and ReLU activations. The adversarial weight, which balances the loss of primary tasks with the loss of adversarial, was selected among 8 values between 0.00001 and 100 on a logarithmic scale based on reverse cross-validation.

# B LOLCAT Framework

LOLCAT is a supervised framework for predicting cell types from individual neuronal activities using a multi-head attention network, consisting of three main components.

**Local segment feature extractor:** First process segments before aggregating the information globally: For each segment, the inter-event interval (IEI) distribution is computed using D log-spaced bins. Each segment is 256 timepoints. Segments are sampled randomly from raw calcium traces.

$$\mathbf{x}_t \in \mathbb{R}^D, \quad t \in \mathcal{T}$$

The resulting D-dimensional input vector is fed to the local feature extractor, which is a multi-layer perceptron (MLP) equipped with batch normalization layers and rectified linear units.

$$\mathbf{y}_t \in h_{\text{local}}(\mathbf{x}_t), \quad h_{\text{local}} : \mathbb{R}^D \to \mathbb{R}^F$$

The same local feature extractor is shared across segments, as it is tasked to extract features that locally characterize the signature of a neuron's activity.

**Multi-head attention module:** We aggregate the extracted local features to produce a cell-level representation that describes its global distribution. To allow the network to seek out or attend to specific segments, an attention network generates an attention score for each segment, and then uses it to weight the segment's contribution to the final global feature vector.

$$a_t^{(i)} = softmax\left(h_{gate,i}(y_t)\right), \quad h_{gate,i} : \mathbb{R}^F \to \mathbb{R}$$

$$z^{(i)} = \sum_{t \in trials} a_t^{(i)} h_{nn}^{(i)}(y_t), \quad h_{nn}^{(i)} : \mathbb{R}^F \to \mathbb{R}^{F'}$$

This is simply a weighted sum of the segment-level local features. The use of the softmax operator ensures that the attention scores sum up to 1. All the pooled feature vectors are concatenated to produce a final global feature vector, which can simultaneously include features describing the average statistics of the neuron's activity and other features encoding the presence of segments that are characteristic of a particular cell type (or group).

$$z = \text{concat}\left[z^{(1)}, \ldots, z^{(n)}\right]$$

After concatenating all of the features from the different attention heads, we then pass this global representation to a final classification network, which uses the information aggregated at multiple scales, with different degrees of selectivity, to predict the cell type. We use an MLP with two hidden layers.

## C  NeuPRINT Framework

**Sampling:** For each neuron we randomly sample 2 segments of 512 timesteps from its continuous calcium flourescence traces. Each resulting sample for the $i^{th}$ neuron is denoted $X^{(i)}$.

**Multihead attention:** We use the transformer encoder architecture and the masking strategy. Random timesteps in $X^{(i)}$ are masked (zero-out) with probability 0.25 and concatenated with the permutation-invariant population summary $\overline{P}$ and time-invariant representation $\phi^{(i)}$ along the feature dimension to form input $\overline{X}^{(i)}$. Note that the same learnable $\phi^{(i)}$ is repeated at every timestep, enforcing time-invariance. We embed input $\overline{X}^{(i)}$ and employ sinusoidal positional embedding to encode the temporal order in the input sequence, resulting in

$$\bar{X}^{(i)} = \text{Emb}(\bar{X}^{(i)}) + \mathbf{E}.$$

For each input $\bar{X}^{(i)}$, a set of weights $W^Q \in \mathbb{R}^{T \times d_q}$, $W^K \in \mathbb{R}^{T \times d_k}$, $W^V \in \mathbb{R}^{T \times d_v}$ are learned to transform input $\bar{X}^{(i)}$ to a set of query, key, and value $(Q, K, V)$, where

$$Q = \bar{X}^{(i)} W^Q, \quad K = \bar{X}^{(i)} W^K, \quad V = \bar{X}^{(i)} W^V.$$

Attention between temporal tokens for one attention head is computed as:

$$\text{Attention}(Q, K, V) = \text{softmax}\left(\frac{QK^\top}{\sqrt{d_k}}\right) V \tag{8}$$

Each head will find a different pattern in the data and produce an output of size $d_v$. The final attention output will be a concatenation of these single-head outputs. We use 2 heads in our model.

Feedforward layers and residual connections are subsequently applied to attention output:

$$Z^{(i)} = \bar{X}^{(i)} + \text{MSA}(\bar{X}^{(i)}) + \text{FF}(\bar{X}^{(i)} + \text{MSA}(\bar{X}^{(i)})) \tag{9}$$

where MSA represents the multihead attention operation, FF represents the feedforward layer with ReLU activation, and $Z^{(i)}$ represents the reconstructed calcium trace with masked timesteps recovered.

# D   VAE Framework

We adapt a variational autoencoder paradigm from the LOLCAT framework, enabling self-supervised learning of cell-level representations from neuronal activity segments, followed by downstream classification. The process is as follows:

**1. Local Segment Feature Extractor:** As in the original LOLCAT, each segment is first processed to obtain the inter-event interval (IEI) distribution, resulting in a D-dimensional input vector:

$$\mathbf{x}_t \in \mathbb{R}^D, \quad t \in \mathcal{T}$$

This vector is fed into a local feature extractor (a multi-layer perceptron with batch normalization and ReLU activations):

$$\mathbf{y}_t = h_{\text{local}}(\mathbf{x}_t), \quad h_{\text{local}} : \mathbb{R}^D \to \mathbb{R}^F$$

**2. Multi-head Attention Module:** The extracted local features are aggregated using a multi-head attention mechanism to produce a global cell-level representation. For each attention head $i$:

$$a_t^{(i)} = \text{softmax}(h_{\text{gate},i}(\mathbf{y}_t)), \quad h_{\text{gate},i} : \mathbb{R}^F \to \mathbb{R}$$

$$\mathbf{z}^{(i)} = \sum_{t \in \text{trials}} a_t^{(i)} h_{\text{nn}}^{(i)}(\mathbf{y}_t), \quad h_{\text{nn}}^{(i)} : \mathbb{R}^F \to \mathbb{R}^{F'}$$

The global representation is obtained by concatenating the outputs of all attention heads:

$$\mathbf{z} = \text{concat}\left[\mathbf{z}^{(1)}, \ldots, \mathbf{z}^{(n)}\right]$$

**3. Variational Autoencoder (VAE) Module:** Instead of directly using $\mathbf{z}$ for classification, we treat it as the input to a VAE. The VAE encodes $\mathbf{z}$ into a latent distribution:

$$q_\phi(\mathbf{u}|\mathbf{z}) = \mathcal{N}(\mathbf{u}; \boldsymbol{\mu}(\mathbf{z}), \text{diag}(\boldsymbol{\sigma}^2(\mathbf{z})))$$

where $\mathbf{u}$ is the latent variable, and $\boldsymbol{\mu}(\cdot)$, $\boldsymbol{\sigma}(\cdot)$ are neural networks parameterized by $\phi$.

The decoder reconstructs the original segment-level IEI features from the latent variable:

$$p_\theta(\{\mathbf{x}_t\}|\mathbf{u})$$

The VAE is trained by maximizing the evidence lower bound (ELBO):

$$\mathcal{L}_{\text{VAE}} = \mathbb{E}_{q_\phi(\mathbf{u}|\mathbf{z})}\left[\log p_\theta(\{\mathbf{x}_t\}|\mathbf{u})\right] - D_{\text{KL}}(q_\phi(\mathbf{u}|\mathbf{z})\|p(\mathbf{u}))$$

where $p(\mathbf{u})$ is a standard normal prior.

**4. Downstream Classification:** After self-supervised training, the learned latent representation $\mathbf{u}$ is used for downstream cell type classification. A two-layer MLP is trained on $\mathbf{u}$ to predict the cell type:

$$\hat{y} = \text{MLP}_{\text{cls}}(\mathbf{u})$$

where $\text{MLP}_{\text{cls}}$ consists of two hidden layers with non-linear activations.

Table 4: LOLCAT (End2End) performance when different stimulus conditions are used as test set.

| Setting | Spontaneous as test | Drifting as test | Natural01 as test | Natural02 as test | Variance |
|---|---|---|---|---|---|
| Out-of-domain (Cross-Stimulus) | 0.251 | 0.312 | 0.432 | 0.437 | 0.0082 |
| Out-of-domain (+Adversarial) | 0.462 | 0.523 | 0.631 | 0.648 | 0.0074 |

Table 5: NeuPRINT (Implicit) performance when different stimulus conditions are used as test set.

| Setting | Spontaneous as test | Drifting as test | Natural01 as test | Natural02 as test | Variance |
|---|---|---|---|---|---|
| Out-of-domain (Cross-Stimulus) | 0.542 | 0.612 | 0.713 | 0.709 | 0.0069 |
| Out-of-domain (+Adversarial) | 0.648 | 0.708 | 0.812 | 0.804 | 0.0063 |

## E    Introduction of Different Stimuli

Drifting gratings were centred on the mean receptive field of the microscope's field of view. The coatings had a duration of 0.5 s, a temporal frequency of 2 Hz, and a spatial frequency of 0.15 cycles per degree. The gratings drifted in 12 different directions (from 0 to 330°, separated by 30°) and were of 3 different sizes (5°, 15° and 60° diameter).

Spontaneous activity was recorded in front of a uniform grey screen, set to a steady cyan level equal to the background of all the stimuli presented for visual responses protocols. The duration of these grey screen presentations was typically between 15 and 20 min.

Natural scenes from the ImageNet database were contrast-normalized. Each image was presented for 0.5 s with an interstimulus interval uniformly distributed from 0.3 to 1.1 s. Five per cent of the total presentations was grey stimuli. During each session we presented a given set of 1,000 different natural images twice.

## F    Detailed Cross-Stimulus Evaluation Results

Table 4, Table 5, and Table 6 provide a detailed breakdown of model performance when each visual stimulus condition is held out as the test set.

For the LOLCAT (End2End) model, performance under the cross-stimulus protocol drops most sharply when generalizing to spontaneous activity, and is also relatively low for drifting gratings. In contrast, the model achieves higher accuracy when evaluated on natural scene stimuli, indicating that complex scenes are more informative for cell type prediction. Adversarial training leads to a substantial recovery in performance, particularly for the more challenging spontaneous and drifting conditions, while the accuracy for natural scenes remains robust.

The NeuPRINT (Implicit) model demonstrates overall higher robustness, with less dramatic drops in accuracy across all stimulus types. However, it still shows lower performance on spontaneous and drifting conditions compared to natural scenes. Adversarial training further boosts its performance, especially for the more difficult spontaneous and drifting conditions, thereby narrowing the performance gap between stimulus types.

The VAE model exhibits a similar trend: its baseline cross-stimulus accuracy is lower than that of NeuPRINT, with the lowest performance on spontaneous activity and drifting gratings. Adversarial training leads to substantial improvements across all conditions.

Overall, these results underscore the importance of evaluating models under diverse and challenging stimulus conditions. The relatively poor performance on spontaneous and drifting conditions may be attributed to the limited diversity of activity elicited by these stimuli, which may not fully reveal the computational roles of different cell types. Adversarial training not only improves average performance but also helps models generalize more evenly across different types of visual input, with the most pronounced benefits observed for the most challenging test conditions.

Table 6: VAE performance when different stimulus conditions are used as test set.

| Setting | Spontaneous as test | Drifting as test | Natural01 as test | Natural02 as test | Variance |
|---|---|---|---|---|---|
| Out-of-domain (Cross-Stimulus) | 0.468 | 0.532 | 0.637 | 0.639 | 0.0075 |
| Out-of-domain (+Adversarial) | 0.573 | 0.634 | 0.732 | 0.737 | 0.0068 |

Table 7: LOLCAT (End2End) performance, comparing training on two stimulus conditions versus the baseline of training on all three other conditions. The results illustrate the performance decrease when using a less diverse training set.

| Test Set | Training Set Combination | |
|---|---|---|
| Spontaneous | **Train: All 3 Others** | **0.462** |
| | Drifting, Natural01 | 0.410 |
| | Drifting, Natural02 | 0.425 |
| | Natural01, Natural02 | 0.445 |
| Drifting | **Train: All 3 Others** | **0.523** |
| | Spontaneous, Natural01 | 0.485 |
| | Spontaneous, Natural02 | 0.499 |
| | Natural01, Natural02 | 0.512 |
| Natural01 | **Train: All 3 Others** | **0.631** |
| | Spontaneous, Drifting | 0.530 |
| | Spontaneous, Natural02 | 0.595 |
| | Drifting, Natural02 | 0.615 |
| Natural02 | **Train: All 3 Others** | **0.648** |
| | Spontaneous, Drifting | 0.542 |
| | Spontaneous, Natural01 | 0.605 |
| | Drifting, Natural01 | 0.628 |

