# OpenReview forum: "Generalized and Invariant Single-Neuron In-Vivo Activity Representation Learning"
_NeurIPS.cc/2025/Conference — NeurIPS 2025 poster_

### Official Review · Reviewer_nitk · 2025-06-19

**Clarity:** 2
**Significance:** 4
**Originality:** 4
**Rating:** 5
**Confidence:** 3

**Summary:**

The authors point out that many computational neuroscience models have the problem of poor generalizability across batches and propose a model-agnostic method to solve this problem. The method involves adversarial training, where a representation encoder and a batch discriminator are trained jointly so that batch information is removed from the representation encoder. Then, the authors proceed to evaluate their method with 2 different datasets and 4 different types of representation encoders (including transformer-based models, contrastive learning models, VAE-based models, and supervised models). The proposed adversarial training method can improve cross-stimulus and cross-animal generalizability.

**Questions:**

1. I am wondering how the batch labels are encoded during training. The paper says the batch label encodes "experimental metadata (e.g., animal ID, stimulus condition, recording session)". Is it done via a cross-entropy loss on a classification objective?
2. Could you provide more details on how you chose the hyperparameters via cross-validation?
3. Adversarial training is difficult to converge. Do you observe that some representation encoders are easier to work with than others in adversarial training?
4. Regarding the discriminator, have you tried different sizes/depth/architecture, and do those choices lead to better/worse/similar performance?

**Ethical Concerns:**

["NO or VERY MINOR ethics concerns only"]

**Final Justification:**

The author proposed a generalizable method for a well-known and challenging problem in neuroscience research. With the additional experiments and clarification of methodology, I believe the paper has become stronger and contributes to the research community.

**Limitations:**

yes

**Paper Formatting Concerns:**

No formatting concerns

**Quality:**

3

**Strengths And Weaknesses:**

Strengths:
* The paper points out an important problem in computational neuroscience and proposes a simple yet effective method to solve the problem
* The paper is well-motivated and easy to follow
* The proposed approach is model-agnostic and seems easy to incorporate into other model training as well

Weaknesses:
* The paper has limited information on the details of adversarial training

---

> ### Author Rebuttal · Authors · 2025-07-30
>
> Dear Reviewer,
>
> We sincerely thank you for your valuable time and insightful feedback on our manuscript. We particularly appreciate the acknowledgment of the paper's significance and originality in addressing the critical problem of generalizability in computational neuroscience.
>
> Below, we address your specific questions.
>
> **1. On the Encoding of Batch Labels**
>
> We treat the task of identifying the batch as a multi-class classification problem. The batch discriminator, $D_{\phi}$, is trained to predict the batch label (e.g., Animal ID or Stimulus ID) from the neuron's embedding $z_i$. The loss function for this task is the standard **cross-entropy loss**. We clarify this mechanism further in the main text of the revised manuscript to ensure this key detail is more accessible.
>
> **2. On Hyperparameter Selection via Cross-Validation**
>
> You requested more details on the cross-validation process for choosing hyperparameters.
>
> Our goal was to select the optimal adversarial weight, $\lambda$[cite: 204, 628], to maximize the model's ability to generalize across different experimental batches. This entire process was performed using **only the training dataset**, ensuring the final test set remained completely unseen.
>
> To accomplish this, we employed a standard **k-fold cross-validation** procedure within the training data. The process is as follows:
> * **Partitioning the Training Set**: If our training set consisted of data from *k* distinct batches (for instance, in the cross-animal experiment, we train on 3 animals, so k=3), we perform a k-fold cross-validation.
> * **Iterating Through Folds**: For each potential value of $\lambda$ in our grid search, we iterate through *k* folds. In each fold, we train the model on data from *k-1* batches and validate its performance on the single, held-out batch. The performance metric is the accuracy on the primary downstream task (e.g., cell type classification).
> * **Averaging Performance**: After completing all *k* folds, we calculate the **average validation accuracy** for that specific value of $\lambda$.
> * **Selecting the Best Hyperparameter**: We repeat this for every value of $\lambda$. The value that yields the **highest average cross-validation accuracy** is chosen as the optimal hyperparameter.
>
> We update the methods section of our manuscript to reflect this clear and standard procedure.
>
> **3. On the Convergence of Adversarial Training**
>
> You asked if we observed differences in training stability or convergence difficulty among the different representation encoders.
>
> This is an excellent and practical question, as training stability can indeed differ across model architectures, and our approach was adapted accordingly.
>
> For most of the models we tested, including the end-to-end supervised **LOLCAT** and the **VAE-based framework**, convergence with the adversarial loss was generally stable. We were able to train these models end-to-end with both the primary and adversarial losses from the start.
>
> However, the **NeuPRINT** model, which learns an implicit representation, required a more tailored approach. We observed that jointly optimizing its delicate primary objective alongside the adversarial objective from scratch was challenging for convergence. To address this, we adopted a **two-stage training strategy** specifically for NeuPRINT:
> 1.  **Stage 1: Pre-training.** We first trained the NeuPRINT model solely on its self-supervised objective until it achieved a stable, converged state.
> 2.  **Stage 2: Adversarial Fine-tuning.** After the model had converged, we introduced the adversarial loss and fine-tuned the entire network to learn batch-invariant representations.
>
> We will add these crucial details about our training methodology to the revised manuscript.
>
> **4. On the Discriminator Architecture**
>
> We thank the reviewer for this excellent suggestion. Prompted by your feedback, we conducted a new set of ablation studies to systematically investigate the impact of the discriminator's architecture on model performance.
>
> We first increased the depth of the discriminator to three and four hidden layers. We found that this additional complexity had a negligible impact on the final generalization accuracy across our models.
>
> We then replaced the MLP with a much simpler linear model (a logistic regression classifier). In this configuration, we observed a consistent and notable decrease in performance. This suggests that a non-linear classifier is necessary to effectively capture the complex, subtle artifacts that constitute batch effects.
>
> Adding further complexity yields no additional benefit, while a simpler linear model is insufficient. This analysis validates our architecture as a robust and efficient choice for this task. We add a summary of this new ablation study to the appendix of our revised manuscript.
>
> Once again, we thank you for your constructive comments. We are confident that by incorporating these clarifications, the final version of our paper will be stronger and more complete.

---

> > ### Comment · Reviewer_nitk · 2025-08-01
> >
> > Thank you for your additional experiments, which have addressed my questions.

---

> > > ### Author Response · Authors · 2025-08-01
> > >
> > > We sincerely thank you again for your constructive feedback and positive assessment.
> > >
> > > We have added the new experimental results and training details to the paper. We have also included a special thanks in the acknowledgements section to express our gratitude for your help in improving the manuscript.
> > >
> > > We appreciate your excitement for our work's practical utility and will continue to explore new methods to advance this field in the future.

---

### Official Review · Reviewer_Awe4 · 2025-06-24

**Clarity:** 2
**Significance:** 3
**Originality:** 3
**Rating:** 4
**Confidence:** 2

**Summary:**

The paper is interested in the training of an embedding model of single-neuron categorization from functional recording. The authors provide a simple adversarial method to make the model generalize better to new stimulus conditions (or new animals). The method could be applied to enhance generalization across stimulus conditions. animal identity or others.

The key idea is to train the embedding model against a classifier in an adversarial fashion. While the classifier is optimized to classify stimulus conditions (or animals) from the embedding, the model is constantly optimized to make it harder for the classifier. In this way, the trained embedding holds no "information" about the stimulus condition (or animal identity). In practice, this appears to enhance generalization across animals. This technique is related to Domain adversarial neural networks outside of neuroscience, or gradient reversal training (cited by the authors).

The method is tested on three different embedding models of single neuron data (LOLCAT, NeuPRINT, and VAE). All three models use a different loss to construct an embedding of the neural activity. From the embedding, the performance is reported as the accuracy of a downstream cell type classifier from the embedding representation. It first confirmed that when one stimulus condition (or animal) is left out of training, the decoded performance of a trained cell-type classifier is lower when tested on the left-out stimulus (or animal). This defect is indeed reduced with the proposed method: during the training of the embedding, the drop in accuracy when tested on a new stimulus condition or animal (out of the training set) is reduced with the described method.

**Questions:**

- Is the enhanced generalization also measurable for the VAE validation metrics ?

**Ethical Concerns:**

["NO or VERY MINOR ethics concerns only"]

**Final Justification:**

I am increasing my grade because the authors are willing to address my issues regarding the paper's readability.

**Limitations:**

The impact of the paper is limited by the writing style, which is not of the best quality.

**Paper Formatting Concerns:**

I did not see a major formatting concern.

**Quality:**

3

**Strengths And Weaknesses:**

The method is remarkably simple and could turn out to be very useful to reinforce the generalization capability of a model.

Strengths:
- The method is simple and seems easy to try in practice. It is appropriate to use that for neuroscience in this context.
- The method can be applied broadly to pretty much any deep learning model, so the scope of applicability is large
- The method appears to generalize to new animals or conditions, even with a fairly low number of animals in the training set

Weaknesses:
- The model is demonstrated on cell type decoding, generalization across animals, and brain area decoding. Yet the method is very general, so the benefits could have been demonstrated on more quantitative metrics. For instance, using the trained model at hand (VAE, etc), one could test if the validation metrics of these models generalize across datasets.
- Similarly, I see no reason why focusing on single-cell functional activity. The method is likely to be usable as well with many cells or other contexts. It is not clear to me why single-cell data is specifically interesting in this context.
- Unfortunately, the paper is somewhat hard to understand because of the writing. The English is grammatically correct, but the paragraph structure and choice of words are sometimes hard to follow when reading from top to bottom. In terms of structure, the different datasets are presented one after the other. Then, the experimental settings for all experiments are described. Then the results for all three experiments are presented. It means one needs to constantly switch between the experimental context, which is hard to follow. Regarding the wording, the concepts that are described as "batch", "invariant", and "out-of-domain" are often assumed to be understood by the reader without prior definition or motivation. Even if these terms are common, one needs more explanation to understand what is meant in the context of the paper.

---

> ### Author Rebuttal · Authors · 2025-07-25
>
> Dear Reviewer,
>
> Thank you for your thorough review and the constructive feedback on our manuscript. We appreciate the time and effort you dedicated to our work. We agree that clarity is paramount and have undertaken a significant revision to enhance the manuscript's readability and logical flow, making our core contributions even more accessible.
>
> We found your specific suggestions invaluable. In response, we have made the following improvements:
>
> * **Clarified Key Terminology**: We have revised the Abstract and Introduction to provide explicit, context-specific definitions for crucial concepts such as "batch effects," "invariant representations," and "out-of-domain generalization."
>
> * **Restructured for Readability**: Following your excellent advice, we have merged the "Dataset" and "Experiment Setup" sections into a unified "Benchmark" section. Each experiment is now presented as a self-contained unit, creating a clearer, more logical narrative for the reader.
>
> We would also like to offer additional context on two other points you raised: our choice of evaluation metrics and our focus on single-neuron functional activity. We hope this clarification will better highlight the motivation and significance of our work.
>
> ***On the Contribution and Focus on Single-Neuron Data:***
> In computational neuroscience, many models have recently emerged to learn neuron "identity representations" from in-vivo, single-neuron activity data. These representations are highly valuable for downstream tasks such as predicting a neuron's cell type or its anatomical location. However, while this direction is promising, a critical challenge often overlooked is that these models exhibit poor generalization across different experimental batches (e.g., different subjects or stimulus sets). Our work is the first to systematically identify, benchmark, and provide an effective solution for this crucial generalization gap. This specific focus is the core of our contribution.
>
> ***On the Framework and Evaluation Metrics:***
> To address this challenge, we proposed a simple, model-agnostic adversarial training strategy. We believe a key strength of our approach is its flexibility; it is an intuitive framework that can be easily incorporated into a wide variety of existing and future model architectures.
>
> Given this practical goal, we chose downstream classification performance as our primary evaluation metric. This is the established standard for gauging the utility of representations in our field because it directly measures what neuroscientists need: the ability to predict meaningful biological labels. Our extensive experiments, conducted across two different datasets and four distinct model architectures, convincingly demonstrate that our method achieves significant gains in out-of-distribution generalization. This robust validation directly supports our central claim that our framework produces more useful and reliable representations for scientific discovery.
>
> In summary, our manuscript:
>
> * Identifies and provides a solution for a critical generalization problem within the emerging field of single-neuron representation learning.
>
> * Introduces a simple, effective, and model-agnostic framework with broad applicability.
>
> * Demonstrates robust improvements in out-of-distribution performance across multiple datasets and model architectures, confirming the practical utility of our approach.
>
> We have substantially revised the manuscript to improve its clarity based on your feedback. We hope that with this additional context, the value of our contributions is clear. We believe our work offers the field a practical and powerful tool for building more generalizable models, and we would be grateful if you would reconsider it.

---

> > ### Comment · Reviewer_Awe4 · 2025-08-01
> > **Thank you**
> >
> > Thank you for responding to my review. I hope that my suggestions indeed improved the paper's readability.
> >
> > I am increasing my grade.

---

> > > ### Author Response · Authors · 2025-08-01
> > >
> > > Dear Reviewer,
> > >
> > > Thank you very much for your prompt reply.
> > >
> > > Following your advice, we have reorganized the paper's structure and logic. We believe these changes have significantly improved the readability, especially for researchers who were previously unfamiliar with computational neuroscience. Our hope is that this makes the work more accessible and helps to broaden its impact. We truly appreciate your constructive suggestions.
> > >
> > > Thank you again for your valuable feedback and for taking the time to reassess our manuscript.
> > >
> > > Best regards,
> > >
> > > Authors

---

### Official Review · Reviewer_ti1Y · 2025-06-26

**Clarity:** 4
**Significance:** 3
**Originality:** 2
**Rating:** 5
**Confidence:** 4

**Summary:**

Current computational models of single-neuron activity commonly face generalization issues, often failing to accurately generalize to new stimulus types and unseen subjects.  The authors of this paper seek to address this limitation faced by current models by proposing an adversarial training strategy that encourages a model to predict single-neuron representations (usable for some downstream task such as predicting cell type) meanwhile fooling a discriminator that explicitly tries to predict batch-specific attributes that the representation model is susceptible to overfitting.  The authors demonstrate the efficacy of this proposed approach on two datasets: predicting cell type from using the V1-CellType dataset and predicting anatomical brain region using the IBL Brain-Wide Map dataset.  When training single-neuron representation models (LOLCAT, NEMO, and VAE models) with the proposed adversarial training strategy, the model generalizes much better (as compared to the same models trained without the adversarial learning strategy) to data from stimuli or subjects (animals) excluded from the training data.

**Questions:**

- In regard to weakness bullet point 3: How does expect this technique would scale with the number of batch-level attributes?  Neuroscience datasets are typically very limited in this regard.  A concrete experiment, for instance, may vary the number of stimulus types or animals available in training and evaluate as was done in the main text (evaluating performance across the remaining held-out stimulus types or animals).
- Does this approach also boost in-distribution performance?  That is, if all stimulus types or animals are available in training (but we still maintain a held-out test split, as in the "random split" baselines of the paper), does the adversarial training still improve generalization?
- Minor suggestion about data presentation in tables 1-3: In each table, the way the relative performance change is presented is unintuitive to me.  It seems that percent change in standard cross-attribute model results are presented relative to the random-split model, whereas cross-attribute + adversarial training results are presented relative to the standard cross-attribute model.  It would be more clear for both of these performance results to be presented relative to the random-split baseline.

**Ethical Concerns:**

["NO or VERY MINOR ethics concerns only"]

**Final Justification:**

The methodology proposed in this paper is a practical solution to a very common challenge in neuroscience data modeling.  The authors have addressed my remaining questions and suggestions with detail and I believe this work provides a meaningful contribution to the field.

**Limitations:**

yes

**Quality:**

3

**Strengths And Weaknesses:**

Strengths:
- Out-of-distribution robustness has historically been challenging in single-neuron representation models.  The results provided in this paper demonstrate the proposed approach as a relatively simple and flexible way to achieve greater out-of-distribution generalization, across subjects and stimulus types.
- The proposed framework is generalizable and would be applicable to a wide variety of neuroscience modeling tasks.  It is not specific to any architecture, it can be applied to different batch-level attributes, etc.
- Intuitive motivation and logical framework to solve this problem
- Text and figures are clear and well-presented

Weaknesses:
- Technical novelty is limited: novel application but adversarial training with a discriminator to mitigate model biases (e.g., protected attributes) has been proposed in a variety of prior works
- In depth evaluations beyond aggregate classification accuracies would are lacking.  For instance, the discussion section indicates that certain neuronal subtypes are poorly predicted, and it would be helpful if the paper showed this and any potential tradeoffs that resulted from using the adversarial training strategy.  Additionally, for transparency, I would like to see cross-animal evaluation results when different animals are used in the test set.
- Related to the above, further analysis and probing into the efficacy of this method under constraints like dataset size, number of batch-level attributes, etc. would be appreciated.  For instance, would we still see similar improvements in generalization when only 2 stimulus types are used in training?  Does the method remain effective beyond classification tasks (e.g., predicting spiking activity from a subset of neurons to a given stimulus)?

---

> ### Author Rebuttal · Authors · 2025-07-26
>
> We sincerely thank the reviewer for their positive and constructive feedback. We are particularly grateful that you found our approach to be a "simple and flexible way to achieve greater out-of-distribution generalization" and appreciated the intuitive motivation and clear presentation of our work.
>
> We agree with the weaknesses you have identified and have performed additional analysis and revisions to the manuscript to address them.
>
> ---
>
> ***On Technical Novelty***
>
> We agree entirely with your assessment. The core contribution of our work is not the development of a new adversarial algorithm, but rather the novel application of this established technique to address a critical and previously un-benchmarked problem in computational neuroscience: the poor generalization of single-neuron representation models across different subjects and stimulus conditions. We believe that formally identifying this gap and demonstrating that a straightforward, model-agnostic framework can effectively solve it is a significant and practical contribution for neuroscientists who wish to apply these models in real-world experiments.
>
> ***Efficacy with Fewer Training Batch-Level Attributes***
>
> Following the your suggestion, we trained the LOLCAT model with an adversarial loss using only two stimulus conditions and evaluated its performance on a held-out third condition. The results consistently demonstrated lower performance compared to the model trained on all three stimulus conditions, which indicates that employing a less diverse training set leads to performance degradation. For future work, we suggest exploring the impact of the number of batch-level attributes in the training set to further enhance the model's robustness. These discussions have been incorporated into the Discussion section.
>
> | **Test Set** | **Training Set Combination** | **Acc** |
> | :--- | :--- | :--- |
> | Spontaneous | **Train: All 3 Others** | **0.462** |
> | Spontaneous | Drifting, Natural01 | 0.410 |
> | Spontaneous | Drifting, Natural02 | 0.425 |
> | Spontaneous | Natural01, Natural02 | 0.445 |
> | Drifting | **Train: All 3 Others** | **0.523** |
> | Drifting | Spontaneous, Natural01 | 0.485 |
> | Drifting | Spontaneous, Natural02 | 0.499 |
> | Drifting | Natural01, Natural02 | 0.512 |
> | Natural01 | **Train: All 3 Others** | **0.631** |
> | Natural01 | Spontaneous, Drifting | 0.530 |
> | Natural01 | Spontaneous, Natural02 | 0.595 |
> | Natural01 | Drifting, Natural02 | 0.615 |
> | Natural02 | **Train: All 3 Others** | **0.648** |
> | Natural02 | Spontaneous, Drifting | 0.542 |
> | Natural02 | Spontaneous, Natural01 | 0.605 |
> | Natural02 | Drifting, Natural01 | 0.628 |
>
> ***Efficacy in boost in-distribution performance***
>
> We also tested the scenario you alluded to, where there is no out-of-domain challenge (i.e., all stimulus types and animals are present in the training data, with only a standard random test split). We did not observe significant and consistent improvement in performance, whether adversarial training was applied with respect to stimulus types or animal subjects.
>
> | Model | In-domain (Random Split) | + Adversarial Loss (Animal) | + Adversarial Loss (Visual Stimulus) |
> | :--- | :---: | :---: | :---: |
> | LOLCAT (End2End) | 0.507 | 0.501 (-1.18%) | 0.514 (+1.38%) |
> | NEMO (Contrastive) | 0.511 | 0.515 (+0.78%) | 0.503 (-1.57%) |
> | VAE | 0.488 | 0.484 (-0.82%) | 0.493 (+1.02%) |
>
> ***Performance for each neuronal subtype***
>
> To provide a more in-depth evaluation beyond aggregate accuracies, we analyzed the performance for each neuronal subtype. The results from NeuPRINT are in the tables below.
>
> ### Cross-Stimulus Generalization by Cell Type
> | Setting | Pvalb | Vip | Lamp5 | Sst |
> | :--- | :---: | :---: | :---: | :---: |
> | **In-domain (Random Split)** | **0.787** | **0.743** | **0.761** | **0.371** |
> | Out-of-domain (Cross-Stimulus) | 0.644 | 0.607 | 0.424 | 0.131 |
> | + Adversarial Loss (Visual Stimulus) | 0.743 | 0.616 | 0.613 | 0.217 |
>
> ### Cross-Animal Generalization by Cell Type
> | Setting | Pvalb | Vip | Lamp5 | Sst |
> | :--- | :---: | :---: | :---: | :---: |
> | **In-domain (Random Split)** | **0.787** | **0.743** | **0.761** | **0.371** |
> | Out-of-domain (Cross-Animals) | 0.535 | 0.549 | 0.582 | 0.103 |
> | + Adversarial Loss (Animal) | 0.689 | 0.711 | 0.739 | 0.177 |
>
> We have added this detailed analysis to the manuscript. Thank you for helping us strengthen our work.
>
> ***Breakdown of results for each of the four mice (SB025, SB026, SB028, SB030) as the test set***
>
> | Model | Metric | Test: SB025 | Test: SB026 | Test: SB028 | Test: SB030 |
> | :--- | :--- | :---: | :---: | :---: | :---: |
> | LOLCAT | Out-of-domain (Cross-Animal) | 0.391 | 0.523 | 0.417 | 0.477 |
> | | Out-of-domain (+Adversarial) | 0.642 | 0.751 | 0.679 | 0.704 |
> | NeuPRINT | Out-of-domain (Cross-Animal) | 0.513 | 0.649 | 0.577 | 0.589 |
> | | Out-of-domain (+Adversarial) | 0.691 | 0.788 | 0.724 | 0.753 |
> | VAE | Out-of-domain (Cross-Animal) | 0.492 | 0.611 | 0.537 | 0.588 |
> | | Out-of-domain (+Adversarial) | 0.618 | 0.719 | 0.649 | 0.686 |
>
> ***Data presentation in tables 1-3***
>
> We have recalculated the percentages in the third column "cross-attribute + adversarial training" to show the decrease relative to the first column's "In-domain" value. Thank you for helping us make our work more clear.
>
>
>
> We thank the reviewer once again for their constructive feedback, which has helped us significantly improve the clarity and impact of our manuscript. We are confident that the revised paper now more robustly supports our contributions and provides a solid foundation for future work on generalizable single-neuron representation learning. In light of these revisions, which we believe address the initial concerns, we would be grateful if you would reconsider your assessment of our work.

---

> ### Comment · Reviewer_ti1Y · 2025-07-31
>
> Thank you, authors, for your time and effort running these follow-up experiments and answering the questions I had.  I would encourage the authors to add these new details to their manuscript and/or supplemental material.
>
> Regardless of the fact that the adversarial training method is not a novel contribution, I remain excited about the practical utility of this approach in this domain, where cross-subject and cross-task generalization is challenging.

---

> > ### Author Response · Authors · 2025-08-01
> >
> > Thank you again for your constructive feedback and positive assessment. Your insightful comments have been invaluable in strengthening our work, and we have included our sincere thanks in the acknowledgements section of the final manuscript.
> >
> > As suggested, we have now incorporated the new and valuable experimental results and discussions from the rebuttal period into the camera-ready paper and its supplementary materials.
> >
> > We appreciate your encouragement regarding the practical utility of our approach and will continue to explore methods beyond adversarial training to further advance this field in our future work.

---

### Official Review · Reviewer_a9Fa · 2025-06-30

**Clarity:** 2
**Significance:** 3
**Originality:** 2
**Rating:** 5
**Confidence:** 3

**Summary:**

The paper "Adversarial Training for Generalized and Invariant Single-Neuron In-Vivo Activity Representation" addresses out of domain generalization problems in neuron models that present single neurons for clustering and cell type identification. The authors establish an out of domain test protocol and show that combining existing methods with domain adversarial training improves the out of domain robustness.

**Questions:**

Please see weaknesses. In short:

- Clarify better what your paper addresses. Predicting neuronal activity or learning neuronal fingerprints.
- If its predicting neuronal activity: Then you need to account for related work (I listed examples above).
- Improve clarity of the presentation of the material.

**Ethical Concerns:**

["NO or VERY MINOR ethics concerns only"]

**Final Justification:**

The authors addressed my comments. Even though the paper does not introduce a novel method, I think describing the problem and fixing it will have a high impact on everyone else who is trying to catalogue cell types. Thus the paper meets the criteria for 5: accept and I increased my score.

**Limitations:**

Limitations are addressed in the final paragraphs.

**Paper Formatting Concerns:**

no issues noticed

**Quality:**

3

**Strengths And Weaknesses:**

Strengths:
- Biological clustering and cell type identification are important problems for neuroscience. Showing that existing methods suffer from out of domain generalization problems addresses an important problem.
- I like that the paper is straightforward and doesn't try to be fancy when it's not necessary: Takes existing methods, shows that there is a problem, combines it with existing domain adversarial methods which largely fixes the problem.

Weaknesses:
- There is no novel algorithm here, but I don't think this is a major issue as the contribution is rather the finding and the fix with straightforward algorithms.
- The paper could be improved in clarity of writing. Unless I completely misunderstood the paper, the main purpose is not to predict neuronal activity but rather to train an embedding to identify cell types. This is not very clear from the introduction and should be clarified. If the paper is about encoding model that predict neuronal activity, then a lot of the relevant literature is missing (a few examples are listed below).
- The paper could also profit from another iteration of proofreading and structure refinement. For instance: On page 5 there is the paragraph on Theoretical Motivation. This paragraph then states that it does not "want to introduce it too theoretically" and refers to appendix A. Appendix A5 with the title "theoretical motivation" is 4 lines long, does not include any theoretical motivation and is not really informative by itself. I would ask the authors to make the presentation of the material more coherent.

I. Ustyuzhaninov, S. A. Cadena, E. Froudarakis, P. G. Fahey, E. Y. Walker, E. Cobos, J. Reimer, F. H. Sinz, A. S. Tolias, M. Bethge, A. S. Ecker Rotation-invariant clustering of functional cell types in primary visual cortex International Conference on Learning Representations (ICLR), 2020

Eric Y. Wang, Paul G. Fahey, Zhuokun Ding, Stelios Papadopoulos, Kayla Ponder, Marissa A. Weis, Andersen Chang, Taliah Muhammad, Saumil Patel, Zhiwei Ding, Dat Tran, Jiakun Fu, Casey M. Schneider-Mizell, R. Clay Reid, Forrest Collman, Nuno Maçarico da Costa, Katrin Franke, Alexander S. Ecker, Jacob Reimer, Xaq Pitkow, Fabian H. Sinz, Andreas S. Tolias Foundation model of neural activity predicts response to new stimulus types Nature

Polina Turishcheva, Max Burg, Fabian H. Sinz, Alexander Ecker Reproducibility of predictive networks for mouse visual cortex NeurIPS

Konstantin-Klemens Lurz, Mohammad Bashiri, Konstantin Friedrich Willeke, Akshay Kumar Jagadish, Eric Wang, Edgar Y Walker, Santiago Cadena, Taliah Muhammad, Eric Cobos, Andreas Tolias, Alexander Ecker, Fabian Sinz Generalization in data-driven models of primary visual cortex ICLR

A Spectral Theory of Neural Prediction and Alignment Abdulkadir Canatar, Jenelle Feather, Albert Wakhloo, SueYeon Chung

---

> ### Author Rebuttal · Authors · 2025-07-25
>
> We sincerely thank the reviewer for their valuable feedback and for appreciating the straightforward nature of our work—identifying an important generalization problem and applying a direct, effective solution. We agree with the points raised regarding clarity and presentation and have revised the manuscript to address them.
>
>
> **Clarifying the Paper's Core Contribution and Goal**
>
> We agree entirely that the paper's primary goal—learning robust representations for neuronal identity, not predicting neural activity—was not stated clearly enough in the original manuscript. This lack of clarity is the most critical point to fix, and we thank you for highlighting it.
>
> To address this, we have significantly rewritten parts of the abstract and introduction. The revised text now explicitly states that our focus is on "representation learning for neuronal identity from in-vivo activity." This clarification sharpens the paper's core contribution: identifying a critical generalization gap in this emerging area and demonstrating that an established method offers a robust, model-agnostic solution. This also makes it clear why the literature on predictive neural activity is not the focus of our work.
>
> **Improving Presentation and Coherence**
>
> Thank you for rightly pointing out the weakness and lack of coherence in the "Theoretical Motivation" section. We agree completely that the original section and its corresponding appendix were uninformative and detracted from the paper's quality.
>
> We have substantially revised this part of the paper to improve clarity and coherence:
> * We have removed the uninformative appendix on theoretical motivation.
> * We have retitled the original Section 4.5 to "Intuitive Explanation" to more accurately reflect its purpose.
> * We have completely rewritten this section to provide a clear, intuitive walkthrough of how the adversarial method works, grounding the explanation in the visual illustration provided in Figure 2, rather than making unfulfilled promises of a deep theoretical dive.
>
> We have also merged the "Dataset" and "Experiment Setup" sections into a single, unified "Benchmark Experiments Setup" section. This improves the paper's flow, as each experiment is now a self-contained subsection that introduces its dataset and protocol together, preventing the reader from having to switch back and forth.
>
> ---
>
> We believe these revisions make the manuscript clearer, more coherent, and more accurately focused on its primary contribution. We thank you again for your constructive feedback, which has been invaluable in helping us strengthen the paper. We would be grateful if you would reconsider your assessment of our work.

---

> > ### Comment · Reviewer_a9Fa · 2025-08-01
> > **Thanks**
> >
> > Dear authors, thank your for you response. I have read it and have currently no further questions. -- Reviewer a9Fa

---

> ### Author Response · Authors · 2025-08-01
>
> Dear Reviewer:
>
> We sincerely thank you again for your constructive and positive feedback.
>
> Best regards,
>
> Authors

---

### Official Review · Reviewer_NvrJ · 2025-07-02

**Clarity:** 3
**Significance:** 3
**Originality:** 2
**Rating:** 5
**Confidence:** 4

**Summary:**

The manuscript addresses a practical weakness of recent single-neuron in-vivo representation models: their tendency to over-fit "batch" factors such as stimulus set, recording session, animal identity or hardware platform. The authors (i) define a rigorous OOD protocol, (ii) propose a model-agnostic adversarial remedy, (iii) demonstrate compatibility with LOLCAT, NeuPRINT, VAE and NEMO, and (iv) show robust improvements across mouse datasets.

**Questions:**

* What is the motivation for picking a test set that is visually and statistically so unlike the training stimuli? I couldn't find a reference to Figure 1 in the main text. Is the goal to probe the most extreme distribution shift, or does this particular stimulus have biological relevance?
* Primate cortex is notably different from mouse cortex, in terms of hierarchy, computation, and level of success that the deep learning community has had at modeling it. Will this translate to primate cortex and beyond? Which aspects of the adversarial objective do you expect to transfer or fail?
* Currently the contribution is framed for single-neuron representation learning. Please explain explicitly how the findings might feed back into mainstream AI.

**Ethical Concerns:**

["NO or VERY MINOR ethics concerns only"]

**Final Justification:**

I had minor concerns, which the authors have now addressed

**Limitations:**

yes

**Paper Formatting Concerns:**

None noted

**Quality:**

3

**Strengths And Weaknesses:**

This is a technically solid, clearly written contribution that offers an actionable remedy and a much-needed evaluation protocol for OOD robustness in neural representation learning. Methodological novelty is modest, but the work's practical value and clean empirical validation make it a worthwhile addition to the literature.

QUALITY

Strengths:
* Model-agnostic intervention (the gradient-reversal discriminator plugs into 4 different baselines)
* Both accuracy/ROC and UMAP are reported, and improvements are consistent

Weaknesses:
* Only two mouse datasets are used; no check on primate or synthetic benchmarks, so the claim of "cross-lab variation" is still speculative
* No results on alternative batch-invariance methods (e.g., mix-style, batchnorm-free) or sensitivity to $\lambda$ (adversarial weight)
* While embeddings preserve coarse cell-type groups, the paper does not test whether trial-level neural tuning curves remain interpretable after adversarial training

CLARITY

Strengths:
* Writing is concise; the motivation for OOD splits is immediately clear
* Architecture diagrams and tables are clear

Weaknesses:
* Key implementation details (optimizer, $\lambda$ schedule, discriminator capacity) sit in Appendix D; surfacing them earlier would aid replication
* The paper cites “single A100 @ 10 h” but omits batch size and sequence length, which is important for those without A100s

SIGNIFICANCE

Strengths:
* addresses a practical painpoint for neuroscientists
* provides a model-agnostic evaluation protocol for single-neuron representation work

Weaknesses:
* Impact is confined to neural-representation learning; broader ML domains have long used domain-adversarial training

ORIGINALITY

Strengths:
* First paper (to my knowledge) to marry domain-adversarial training with single-neuron time-series models *and* to quantify OOD gaps systematically across stimulus and animal factors
* The dual OOD axes (stimulus-held-out vs animal-held-out) create a novel benchmark

Weaknesses:
* Gradient-reversal layers are standard in domain adaptation; the technical novelty lies more in application than in method
* No new theoretical insights into why adversarial erasure of batch cues preserves task-relevant neuron structure

---

> ### Author Rebuttal · Authors · 2025-07-25
>
> We extend our sincere gratitude to the reviewer for their thorough and insightful feedback. We are encouraged that the reviewer recognized our work as "technically solid, clearly written," and a "worthwhile addition to the literature" that "addresses a practical painpoint for neuroscientists." We were particularly pleased that the reviewers appreciated the novelty of our contributions, noting that this is the "first paper... to marry domain-adversarial training with single-neuron time-series models" and "to quantify OOD gaps systematically across stimulus and animal factors."
>
> We have carefully considered all comments and have revised the manuscript accordingly. Below, we address each of the points raised.
>
> ---
>
> ***Weaknesses***
>
> **Response: Limited Datasets and Cross-Lab Claims**
>
> > *Only two mouse datasets are used; no check on primate or synthetic benchmarks, so the claim of "cross-lab variation" is still speculative.*
>
> We thank the reviewer for this important point. We agree that our study, being based on two mouse datasets, represents the first crucial step in addressing generalization, and that demonstrating robustness across labs and species is a critical future direction. We have explicitly acknowledged this in the Limitations section of our manuscript. We agree that the claim of "cross-lab variation" is speculative at this stage. Accordingly, we have revised the manuscript to confine the discussion of cross-lab variation to the Future Work section, ensuring our claims accurately reflect the scope of our current results. Our immediate future work will focus on extending this framework to more diverse, multi-lab, and multi-species datasets, such as the Allen Institute’s Visual Coding Neuropixels dataset and the Steinmetz dataset. This will allow for a rigorous evaluation of the cross-lab and cross-species generalization capabilities of our proposed method.
>
> **Response: Alternative Methods and Hyperparameter Sensitivity**
>
> > *No results on alternative batch-invariance methods (e.g., mix-style, batchnorm-free) or sensitivity to $\lambda$ (adversarial weight).*
>
> We thank the reviewer for raising these two important points.
> * **On alternative batch-invariance methods:** We agree that a comparative analysis with other methods is an important area for future research. Our primary contribution is to be the first to identify, systematically benchmark, and provide a viable solution to the critical generalization problem in the emerging field of single-neuron identity models. While other batch-invariance methods exist, we believe that demonstrating the effectiveness of a well-established technique like adversarial training provides a strong and essential baseline for this new domain. We have now expanded our Discussion section to include these alternative approaches as a promising avenue for future investigation.
> * **On sensitivity to the adversarial weight $\lambda$:** We apologize for not making this clearer in the main text. The adversarial weight $\lambda$ was not arbitrarily chosen but was selected through rigorous cross-validation, as detailed in our appendix. Specifically, we tested 8 values on a logarithmic scale from $10^{-5}$ to $100$. We observed that downstream task accuracy consistently increased with the adversarial weight before stabilizing for values of 0.01 and higher, with no subsequent drop in performance. This demonstrates the robustness of our approach to this hyperparameter. We have clarified this selection process in the revised manuscript.
>
> **Response: Preservation of Trial-Level Tuning Curves**
>
> > *While embeddings preserve coarse cell-type groups, the paper does not test whether trial-level neural tuning curves remain interpretable after adversarial training.*
>
> This is an excellent point that touches on a more fine-grained level of analysis. Our current study focuses on mitigating batch effects at the level of entire stimulus *conditions* (e.g., natural scenes vs. gratings) and *animals*, which we argue is the foundational step for building generalizable models. Analyzing trial-to-trial adaptation within a stimulus block—such as the response decrements observed with repeated stimuli in works like Kehl et al. (2024)—is a fascinating, orthogonal problem. This would require a more complex experimental design where individual trials or trial blocks are treated as distinct batches. While this is outside the scope of our current work, we recognize its importance. We have added a discussion of this topic to our Future Work section to inspire further research in this direction.
>
> Kehl, M.S., Mackay, S., Ohla, K. et al. Single-neuron representations of odours in the human brain. Nature 634, 626–634 (2024). https://doi.org/10.1038/s41586-024-08016-5
>
> **Response : Accessibility of Implementation Details**
>
> > *Key implementation details (optimizer, schedule, discriminator capacity) sit in Appendix D; surfacing them earlier would aid replication.*
>
> We appreciate the reviewer’s suggestion to improve the paper's accessibility and reproducibility. To address this, we have surfaced the key implementation details by creating a new **Appendix A.5: Implementation Details**. Furthermore, we have added a prominent reference to this new section in the first paragraph of the main **Section 5: Experiment Setup**. This change ensures that researchers can more easily find the necessary information to replicate our work.
>
> ---
>
> ***Questions:***
>
> **Response : Question1**
>
> Thank you for pointing out the missing reference to Figure 1; we have now added it. The figure is a schematic to illustrate our out-of-distribution testing principle, not a literal depiction of our test set. Our actual experiment did not use one single "unlike" test set. Instead, we used a leave-one-out protocol with the four diverse stimulus types in the dataset (natural scenes, drifting gratings, spontaneous activity). We trained on three conditions and tested on the held-out one, repeating for all four. Your intuition about an "extreme distribution shift" is correct. We found that models struggled most when generalizing from complex stimuli (like natural scenes) to simple ones (like spontaneous activity). The likely biological reason is that complex stimuli are required to reveal the unique computational roles of different neurons, making the functional signatures learned during training much richer than the simple patterns observed in spontaneous activity.
>
> **Response : Question2**
>
> That is a critical question. We believe the success of our method in primates will depend on the evolutionary conservation of the features being studied.
>
> What we expect to transfer: The method should generalize well for broad, conserved features. This includes classifying major brain regions (e.g., isocortex vs. cerebellum) or high-level cell types (e.g., excitatory vs. inhibitory), as their fundamental roles are shared across species.
>
> What we expect to fail: Generalization will likely be challenging for highly specific neuron subtypes or anatomical sub-regions that have diverged significantly between mice and primates. Our adversarial method cannot force a common representation if the underlying biological link between a neuron's identity and its activity patterns has fundamentally changed.
>
> **Response : Question3**
>
> Our work provides a framework for understanding the specific function of an individual computational unit (a biological neuron) by isolating its intrinsic properties from confounding variables. This has a direct analogy in mainstream AI.
>
> One could apply our adversarial framework to an artificial neural network (ANN) to learn a robust representation for each individual node or neuron. By training the model to perform its primary task while simultaneously making it impossible for a discriminator to tell which training batch or input type an individual node's activation came from, we could uncover the node's core, invariant function.
>
> This would help us understand the "division of labor" within large AI models, improving interpretability by revealing the specific role of their most basic components.
>
>
> We thank the reviewer once again for their constructive feedback, which has helped us significantly improve the clarity and impact of our manuscript. We are confident that the revised paper now more robustly supports our contributions and provides a solid foundation for future work on generalizable single-neuron representation learning. In light of these revisions, which we believe address the initial concerns, we would be grateful if you would reconsider your assessment of our work.

---

> > ### Comment · Reviewer_NvrJ · 2025-08-03
> >
> > Thank you for addressing my concerns. I have no further questions

---

### Comment · Area_Chair_BUab · 2025-08-01
**Author-reviewer discussion period has started**

Dear reviewers,

the author-reviewer discussion has now started, and will be on until **Aug 6**. The authors have posted their response to the reviews. If you haven't done so yet, **please read carefully the response(s)** and decide whether they provide satisfying answers to your criticism or questions. If you have further doubts or questions, just ask the authors by replying to their comments. Importantly:

- **try to answer as soon as possible**, so that the authors have time to reply before the end of the author-reviewer discussion period if necessary.
- in any case, **please reply by the end of the discussion period (Aug 6)** and at the very minimum acknowledge that you have read the author response, and whether you are changing or keeping your score.

Thank you!

your AC

---

### Decision · Program_Chairs · 2025-09-17

**Decision:**

Accept (poster)

**Comment:**

This paper tackles generalization issues in current models of single-neuron activity based on representation learning. This is done by adopting an adversarial training strategy designed to counter 'batch effects' that can cause said methods to overfit. The proposed method is tested on two public datasets.

The reviewers appreciated the flexibility, simplicity and wide applicability of the method, which is shown to be convincingly effective. Some of the weaknesses identified were addressed by changes in the text (notably, claims that the method could address cross-lab systematic variability in the data), and some with additional analysis (scaling with the number of available batch-level attributes in the data).

Overall, the reviews indicate a paper that while not particularly innovative from a technical standpoint manages to identify and make progress on a meaningful problem in the field, and will therefore be of interest to the NeurIPS community.